# Relation of Alcohol Intake to Kidney Function and Mortality Observational, Population-Based, Cohort Study

**DOI:** 10.3390/nu14061297

**Published:** 2022-03-18

**Authors:** Massimo Cirillo, Giancarlo Bilancio, Carmine Secondulfo, Gennaro Iesce, Carmela Ferrara, Oscar Terradura-Vagnarelli, Martino Laurenzi

**Affiliations:** 1Department of Public Health, University of Naples “Federico II”, 80131 Naples, NA, Italy; 2Department “Scuola Medica Salernitana”, University of Salerno, 84081 Baronissi, SA, Italy; giancarlo.bilancio@gmail.com (G.B.); c.secondulfo1@studenti.unisa.it (C.S.); giesce@unisa.it (G.I.); carmen17789@gmail.com (C.F.); 3Centro Studi Epidemiologici di Gubbio (PG), 06024 Gubbio, PG, Italy; oscar.terradura@gmail.com (O.T.-V.); mlaurenzi@comcast.net (M.L.)

**Keywords:** alcohol, wine, eGFR, mortality, epidemiology

## Abstract

Data are conflicting about the effects of alcohol intake on kidney function. This population-based study investigated associations of alcohol intake with kidney function and mortality. The study cohort included adult participants in Exam-1, Exam-2 (6-year follow-up), and Exam-3 (20-year follow-up) of the Gubbio study. Kidney function was evaluated as estimated glomerular filtration rate (eGFR, CKD-Epi equation, mL/min × 1.73 m^2^). Daily habitual alcohol intake was assessed by questionnaires. Wine intake accounted for >94% of total alcohol intake at all exams. Alcohol intake significantly tracked over time (R > 0.66, *p* < 0.001). Alcohol intake distribution was skewed at all exams (skewness > 2) and was divided into four strata for analyses (g/day = 0, 1–24, 25–48, and >48). Strata of alcohol intake differed substantially for lab markers of alcohol intake (*p* < 0.001). In multivariable regression, strata of alcohol intake related cross-sectionally to eGFR at all exams (Exam-1: B = 1.70, *p* < 0.001; Exam-2: B = 1.03, *p* < 0.001; Exam-3: B = 0.55, *p* = 0.010) and related longitudinally to less negative eGFR change from Exam-1 to Exam-2 (B = 0.133, *p* = 0.002) and from Exam-2 to Exam-3 (B = 0.065, *p* = 0.004). In multivariable Cox models, compared to no intake, intakes > 24 g/day were not associated with different mortality while an intake of 1–24 g/day was associated with lower mortality in the whole cohort (HR = 0.77, *p* = 0.003) and in the subgroup with eGFR < 60 mL/min × 1.73 m^2^ (HR = 0.69, *p* = 0.033). These data indicate a positive independent association of alcohol intake with kidney function not due to a mortality-related selection.

## 1. Introduction

Guidelines for nutrition in chronic kidney disease do not include any indication regardingalcohol intake [1]. Nevertheless, a recent metanalysis concluded that, compared with no consumption, moderate alcohol consumption is associated with a reduced risk of chronic kidney disease [2]. At the level of the general population, the possible effects of alcohol intake on glomerular filtration rate are an unanswered question given that epidemiological studies reported either no independent association [3,4,5,6,7,8], negative association [9,10], or positive association [11,12,13,14,15,16]. The inconsistency of the findings could reflect the confounding effect of several factors, including socio-cultural heterogeneity in the definitions of moderate and heavy drinking patterns [17,18], heterogeneity in the target indices of kidney function [19,20], lack of information on other major dietary modulators of kidney function as protein or salt [2], and the cross-cultural heterogeneity in the type of the predominant alcohol-containing beverages (beer, wine, spirits, etc.) [21]. In the light of the above, the countries of the Mediterranean area, and in particular Italy, represent a peculiar model because wine is by far the most important and often exclusive source of habitual alcohol intake [21,22]. Therefore, the present analysis was designed to investigate the relationship of habitual intake of alcohol with kidney function and with its long-term changes over time in a sample of the Italian general population using updated methods of glomerular filtration rate estimation and controlling for dietary and non-dietary correlates of kidney function.

## 2. Materials and Methods

The Gubbio study is a population-based, observational, longitudinal project ongoing in the city of Gubbio, in northern-central Italy [23,24,25]. The study adheres to the Declaration of Helsinki and includes informed consent and approval by the institutional committee (CEAS-Umbria #2850/16). The study also investigates the relationship of dietary factors with renal endpoints [26,27,28,29,30]. The study design, involvement of the invited population, response rates, and characteristics of the Gubbio study cohort are reported elsewhere [23,24,25]. In short, three main exams were performed: the baseline exam in 1983–85 (Exam-1) and two follow-up exams in 1989–92 (Exam-2) and 2001–07 (Exam-3). At all exams, the study protocol included the administration of standardized questionnaires by trained personnel and a medical visit with the measurements of anthropometry and blood pressure. Blood pressure was measured, on the right arm, by trained physicians after participants were seated quietly for 5 min with the use of mercury sphygmomanometers and cuffs of appropriate size. Three measurements were taken one minute apart, and the means of the second and third measurements were used in the analyses.

Regarding biological samples, the protocol for Exam-1 included the collection of a daytime, untimed, spot urine sample and a venous blood sample after a fast of at least two hours [23]. The protocol for Exam-2 included the collection of an overnight timed urine collection under the fed condition from the first void after the evening meal to the first void at the morning wake-up included [26,27]; an early morning blood sample collected under fasting conditions after the completion of the overnight urine collection; a morning timed urine sample under fasting conditions after blood sampling [31]. The protocol for Exam-3 included the collection of an early morning blood sample under fasting conditions. Mortality data were collected after Exam-1 from the local sections of national registries. Individuals with age ≥ 18 years at Exam-1 were the target cohort of the present study.

### 2.1. Variables in Analyses

The analyses of the present study focused on data collected at Exam-1, Exam-2, and Exam-3 together with mortality data from Exam-1 up to the date of the completion of Exam-3 (30 June 2007). Estimated glomerular filtration rate (eGFR) was used as an index of kidney function. Habitual alcohol intake was evaluated at each exam by eight items of the questionnaire separately dealing with the habitual daily consumption of four types of alcohol-containing beverages: wine, beer, aperitifs/cocktails, and spirits/liquors [23,25]. Erythrocyte mean corpuscular volume and gamma-glutamyl-transpeptidase were used as objective markers for the validation of the reported alcohol intake [32]. Variables selected for the analyses were as follows: gender, age, and date of the exam; sodium/creatinine ratio and potassium/creatinine ratio in the Exam-1 daytime urine, and in the Exam-2, overnight urine used as indices of dietary intake of sodium and potassium, respectively [28,29,33]; urea nitrogen/creatinine ratio in the Exam-2 overnight urine used as an index of dietary intake of protein [34]; 24 h urinary creatinine used as an index of creatinine generation and muscle mass [35]; body mass index used as an index of overweight; blood pressure and use of antihypertensive drugs; serum total cholesterol; smoking; serum glucose and use of antidiabetic drugs; date of death. Urinary albumin/creatinine ratio was measured and included in the analyses only for those examinees aged 45–64 years at Exam-2 [36].

### 2.2. Calculations

The follow-up duration for analyses of eGFR was calculated as the time interval between the exams. The follow-up duration for analyses of mortality was calculated as the time interval between Exam-1 and the date of death and as the time interval between Exam-1 and Exam-3 completion in surviving examinees (30 June 2007). eGFR was calculated by the Chronic Kidney Disease–Epidemiology Collaboration equation using gender, age, and serum creatinine [20,37]. The change in eGFR from one exam to the subsequent one was expressed as annualized eGFR change, which is divided by the years of follow-up duration. The eGFR slope per year of follow-up was calculated by regressing the eGFR values of Exam-1, Exam-2, and Exam-3 over the dates of the exams.

The habitual alcohol intake of the four types of alcohol-containing beverages—i.e., wine, beer, aperitifs/cocktails, and spirits—was expressed as g/day using the Italian values of volume and alcoholic graduation of standard servings (Appendix A of Appendix A) [38]. The total habitual alcohol intake as g/day at each exam was calculated as the rounded sum of the habitual intake of alcohol of the four types of beverages.

As for covariates, body mass index was calculated as weight/height^2^; 24 h urinary creatinine was estimated as reported [39]. Diabetes was defined as the use of regular anti-diabetic treatment and/or as serum glucose ≥ 11.1 mmol/L at Exam-1 (blood withdrawal after fast of at least two hours) or ≥ 7.0 mmol/L at Exam-2 and Exam-3 (blood withdrawal after overnight fast).

Serum creatinine was measured in frozen samples by automated biochemistry using a kinetic alkaline picrate assay with IDMS-traceable standardization [19]. The other lab variables were measured in fresh samples using automated biochemistry and quality controls [23].

### 2.3. Statistical Analyses

The study included cross-sectional and longitudinal analyses. The cross-sectional analyses investigated the associations of habitual alcohol intake with lab markers of alcohol intake and eGFR separately at Exam-1, Exam-2, and Exam-3. The longitudinal analyses on eGFR data investigated separately three associations: that of habitual alcohol intake and covariates at Exam-1 with the eGFR change from Exam-1 to Exam-2; that of habitual alcohol intake and covariates at Exam-2 with the eGFR change from Exam-2 to Exam-3; finally, that of the mean of alcohol intake and covariates during the follow-up with the eGFR slope over time from Exam-1 to Exam-3. The mean alcohol intake during follow-up was calculated as the mean of total alcohol intake at Exam-1, Exam-2, and Exam-3. The mean values of covariates during follow-up were calculated as the mean of data available from Exam-1 to Exam-3. The last set of longitudinal analyses investigated the association of alcohol intake and covariates at Exam-1 with a mortality rate as a possible selection bias due to alcohol-related mortality [40].

For statistical analyses, alcohol intake was divided into four strata: intake = 0 g/day, intake in the range 1–24 g/day, intake in the range 25–48 g/day, and intake >48 g/day. ANOVA was used for investigation on the associations of the stratum of alcohol intake with lab markers of alcohol intake and with eGFR data (eGFR, eGFR change over time, and eGFR slope over time). These associations with eGFR data were also investigated using multivariable linear regression where the four strata of alcohol intake were entered as separate 1/0 dummy variables, and the stratum with 0 intake was used as reference. Multivariable Cox models were used to analyze the relationship of the stratum of alcohol intake at Exam-1 with mortality. The covariates used for multivariable analyses are listed in Tables’ footer or Figures’ legend.

Descriptive data were reported as prevalence for categorical variables, mean ± standard deviation (SD) for non-skewed numerical variables, and median with interquartile range (IQR) for skewed numerical variables (skewness > 1). Skewed variables were log-transformed for regression analyses. Results of linear regression were reported as regression coefficient (B) and as standardized regression coefficient (beta) for direct comparability among different variables and among different models. Results were reported, including the 95% confidence interval (95%CI). Statistical procedures were performed using IBM-SPSS Statistics 19 software (IBM, Armonk, NY, USA).

## 3. Results

### 3.1. Descriptive Statistics

Figure 1 show the number of participants with age ≥ 18 years and complete data at Exam-1, examinees lost to follow-up after Exam-1, examinees who died before undergoing Exam-2, examinees participating in Exam-2, examinees lost to follow-up after Exam-2, examinees who died before undergoing Exam-3, and examinees participating in Exam-3. The mortality-corrected response rate was 75.7% at Exam-2 and 82.5% at Exam-3. Of the 2075 participants in all exams, 6 were excluded due to missing data at Exam-2 and/or Exam-3. The 2069 examinees with complete data at all exams made up the study cohort for the analyses on lab markers of alcohol intake and eGFR, while the 4524 examinees with age ≥ 18 years and complete data at Exam-1 made up the study cohort for the analysis on mortality.

Table 1 report descriptive statistics on sex, age, alcohol intake, eGFR, and covariates in the 2069 examinees with complete data at the three exams. Urinary albumin/creatinine ratio was measured only at Exam-2 in examinees with age 45–64 years (mg/g: median = 6.0, IQR = 3.4/11.8).

Age and eGFR data were not skewed (skewness < 0.6). Alcohol intake was positively skewed at all exams (Figure 2). The median alcohol intake was similar in the three exams, while the IQR range was larger at Exam-2 (Table 1).

The Pearson correlation coefficient of alcohol intake was 0.686 between Exam-1 and Exam-2 and was 0.664 between Exam-2 and Exam-3. In alcohol-drinkers, the intake of alcohol in the form of wine accounted for 97.1% of total alcohol intake at Exam-1, for 98.8% at Exam-2, and for 94.7% at Exam-3. At all exams, there were robust trends of the relationship between reported alcohol intake and erythrocytic mean corpuscular volume (Figure 3). A similar trend was found with gamma-glutamyl transferase at Exam-2 (Figure 4).

At all exams, male sex, age, urinary creatinine, body mass index, and systolic pressure correlated positively with alcohol intake (Appendix A). Alcohol intake was associated positively with systolic pressure but not with antihypertensive drug treatment at all exams (Appendix A).

Descriptive statistics in Appendix A report data at Exam-1 in examinees participating in all exams and examinees with missing Exam-2 or Exam-3 because of death during follow-up or loss to follow-up. Regarding alcohol intake, the differences between the group of examinees with all exams and the group of examinees with missing exams were ≤ 2.0% and inconsistent: examinees with all exams had a 2.0% higher prevalence of no alcohol intake (29.0% and 27.0%), 2.0% lower prevalence of alcohol intake 1–24 g/day (43.5% and 45.5%), 1.9% lower prevalence of alcohol intake of 25–48 g/day (11.1% and 13.0%), but 1.9% higher prevalence of alcohol intake > 48 g/day (16.4% and 14.5%).

The means of annualized eGFR change and eGFR slope were negative. Annualized eGFR change from Exam-2 to Exam-3 was 1.67-time greater in comparison to annualized eGFR change from Exam-1 to Exam-2 (Table 2).

### 3.2. Cross-Sectional Analyses on eGFR

At all exams, eGFR differed among strata of alcohol intake (Figure 5, ANOVA without adjustment and with adjustment for covariates). A positive linear trend of eGFR along alcohol strata was significant in multivariable regression at all exams (Exam-1: B= 1.70, 95%CI = 1.00/2.40, *p*< 0.001; Exam-2: B = 1.03, 95%CI = 0.59/1.48, *p* < 0.001; Exam-3: B= 0.55, 95%CI = 0.13/0.98, *p* = 0.010). At Exam-2, the trend was identical also when controlling for log-transformed urinary/albumin ratio in the subgroup with measured urinary albumin (*n* = 956, age = 45–64 years, B = 1.02, 95%CI = 0.38/1.66, *p*= 0.002).

Compared to no intake, alcohol intake in the range 25–48 g/day was found to be related to higher eGFR both at Exam-1 and at Exam-2, while alcohol intake > 48 g/day related to higher eGFR at all exams (Table 3).

Findings were similar when the multivariable model for Exam-1 was analyzed in the examinees who did not participate in follow-up exams (Appendix A). Covariates independently associated with eGFR at all exams were sex, age, urinary creatinine, body mass index, and serum total cholesterol (Appendix A).

### 3.3. Longitudinal Analyses on eGFR

Annualized eGFR changes differed among strata of alcohol intake either in the follow-up from Exam-1 to Exam-2 and in the follow-up from Exam-2 to Exam-3 in ANOVA without adjustment and with adjustment for covariates (Figure 6, upper and intermediate panels). A positive linear trend of eGFR change along alcohol strata was significant in the multivariable regression for follow-up from Exam-1 to Exam-2 (B = 0.133, 95%CI = 0.049/0.216, *p* = 0.002) and for follow-up from Exam-2 to Exam-3 (B = 0.065, 95%CI = 0.024/0.111, *p* = 0.004). The trend for follow-up from Exam-2 to Exam-3 was similar when also controlling for log-transformed urinary/albumin ratio in the subgroup with measured urinary albumin (*n* = 956, age = 45–64 years, B = 0.063, 95%CI = 0.013/0.132, *p* = 0.035).

In multivariable regression (Table 4), compared to no intake, alcohol intake > 48 g/day related to less negative eGFR change from Exam-1 to Exam-2 and from Exam-2 to Exam-3. Covariates associated with eGFR changes in both follow-up periods were sex, age, eGFR, and urinary creatinine at the initiation of the follow-up period (Appendix A).

The eGFR slope over time from Exam-1 to Exam-3 differed among strata of mean alcohol intake in ANOVA without adjustment and with adjustment for covariates (Figure 6, lower panel). A positive linear trend to less negative eGFR slope along strata of mean alcohol intake was significant in multi-variable regression (B = 0.044, 95%CI = 0.016/0.072, *p* = 0.002). The trend was similar when the regression was analyzed separately in the subgroup with Exam-1 systolic pressure ≤ the median (B = 0.049, 95%CI = 0.011/0.086, *p* = 0.012) and in the subgroup with Exam-1 systolic pressure > the median (B = 0.039, 95%CI = −0.003/0.080, *p* = 0.066). In multivariable regression (Table 3), compared to no intake, alcohol intake > 48 g/day related to less negative eGFR slope (Table 3). Covariates associated with eGFR slope were sex and age, eGFR, and urinary creatinine at Exam-1 (Appendix A).

### 3.4. Analysis of Mortality

In multivariable Cox regression models (Table 5), compared to no intake, and alcohol intake at Exam-1 in the range 1–24 g/day associated with 23.3% lower mortality rate independently of sex, age, and other covariates. An alcohol intake in the range 25–48 g/day and in the range > 48 g/day was not associated with a different mortality rate. Table 4 show that findings were similar when the model was analyzed in the subgroup with eGFR in the range 89–60 mL/min × 1.73 m^2^ and in the subgroup with eGFR < 60 mL/min × 1.73 m^2^. Alcohol intake in the range 1–24 g/day did not relate to lower mortality when the regression was re-run excluding the 1491 examinees with alcohol intake > 0 at Exam-1 who moved to the group with no alcohol intake at Exam-2 (HR = 0.973, 95%CI = 0.79/1.20, *p* = 0.797).

## 4. Discussion

The results of the present long-term observational study indicate that, in an Italian sample of the general population with adult age, a higher habitual alcohol intake relates cross-sectionally to a higher eGFR and longitudinally to a less negative long-term eGFR change over time independently of gender, age, and several other variables. Moreover, the study reports the novel observation that moderate alcohol intake in the range of 1–24 g/day is also associated with reduced mortality in the individuals with decreased eGFR.

The main limitation of the study includes the fact that data collection was not recent. However, the importance of this limitation is reduced by the consistency of the findings in three separate exams performed over a 20-year observation period from 1983–85 (Exam-1) to 2001–07 (Exam-3). Another limitation is the fact that the study did not collect complete information on the habitual diet composition but only data about the dietary intake of sodium, potassium, and protein. However, only 2 of the 14 previous studies on alcohol and kidney function reported data about diet composition [3,4,5,6,7,8,9,10,11,12,13,14,15,16]. The confounding should have been minor for examinees dead during follow-up or lost to follow-up because the differences in alcohol intake were minor and inconsistent in comparisons to examinees with complete data at all exams. The last limitation could be the fact that the study can give information on alcohol intake in the form of wine but not of other beverages, given that wine accounted for more than 94% of total alcohol intake at all exams in the Gubbio population. This peculiarity was expected in an Italian population sample [21,22] and could also be considered a merit of the study because the homogeneity of the drinking patterns of the Gubbio cohort implies a lower confounding effect due to differences between wine and other beverages. The merits of this study are the use of updated methods for accurate eGFR calculation, the analyses of datasets from three separate exams, the analyses of erythrocytic mean corpuscular cell volume and serum gamma-glutamyl transpeptidase as objective markers of alcohol intake, and the inclusion in analyses of mortality as a potential determinant of selection-bias due to alcohol-related mortality [40].

As for the inconsistency of the findings between the present study and the previous studies, alcohol intake did not associate with the change of kidney function over time in five studies (3–8). Important methodological differences could explain the lack of significant associations in those previous studies. Two of them were based on eGFR calculation with low accuracy in the normal range of kidney function (3,4); three studies did not focus on eGFR change but only on the incidence of eGFR <60 mL/min × 1.73 [5,7,8]; two studies included the elderly only [6,7]. In contrast with the present results, two studies reported detrimental effects of alcohol intake on kidney function [9,10]. However, they could not be considered relevant to the present study because they investigated only heavy alcohol drinking and alcohol abuse or alcohol dependence [9,10]. The remaining studies on alcohol and kidney function reported favorable effects of alcohol intake on kidney function in accordance with the present results [11,12,13,14,15,16,17].

The mechanisms remain hypothetical for the relationship of higher alcohol intake in the form of wine with higher eGFR and less negative eGFR change over time. Theoretically, alcohol in the form of wine could positively affect kidney function due to its content of polyphenols [41], via the inhibition of antidiuretic hormone secretion [42,43], or via its effects on the endothelium [44]. Other mechanisms cannot be excluded.

Regarding practical implications, this study supports the idea that there is no need to forbid the intake of wine in people with or at risk of low kidney function. This idea is further supported by the evidence that an alcohol intake of 1–24 g/day, corresponding to an intake of wine up to two glasses per day, is associated with a 23% reduced mortality rate in keeping with the so-called “French paradox” [45]. The possibility that this finding reflected only higher mortality in the stratum with no alcohol intake could not be excluded because the results were not consistent in the subgroup analysis excluding examinees who moved to the stratum with no alcohol intake only after Exam-2. In this light, two useful novel observations were reported by the present study: first, the reduction in mortality rate associated with moderate wine intake was detectable also in individuals with decreased eGFR; second, higher wine intake was associated cross-sectionally and longitudinally with better kidney function but not with increased mortality risk.

## 5. Conclusions

Briefly, this observational cohort study reports that, in an Italian sample of the adult general population, independent of gender, age, and several other variables, a higher alcohol intake in the form of wine was related cross-sectionally to a higher eGFR and longitudinally to a lesser eGFR decline during an observation period ranging from 6 to 20 years. Altogether, these results support the hypothesis that the intake of wine could have favorable effects against the decline in kidney function associated with ageing without implying an increased rate of mortality.

## Figures and Tables

**Figure 1 nutrients-14-01297-f001:**
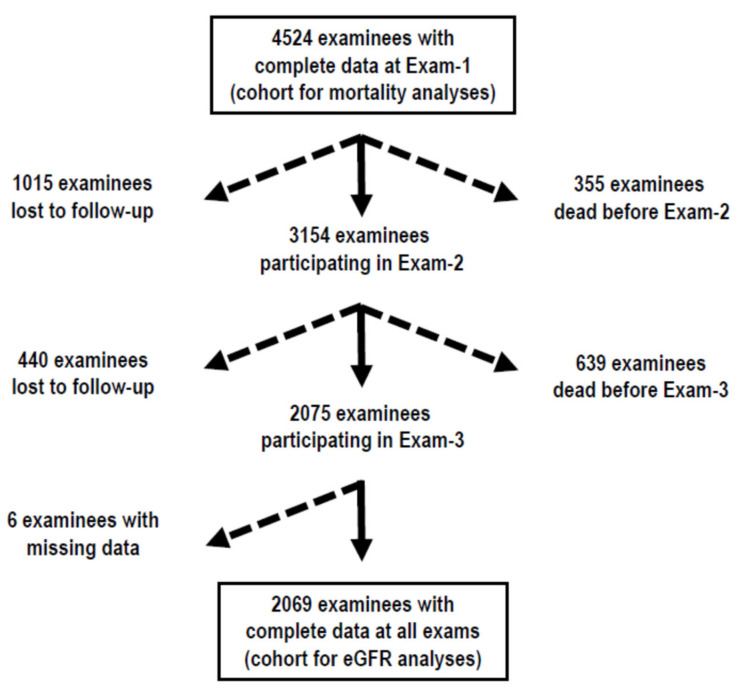
Examinees participating in each exam, examinees lost to follow-up, and dead examinees.

**Figure 2 nutrients-14-01297-f002:**
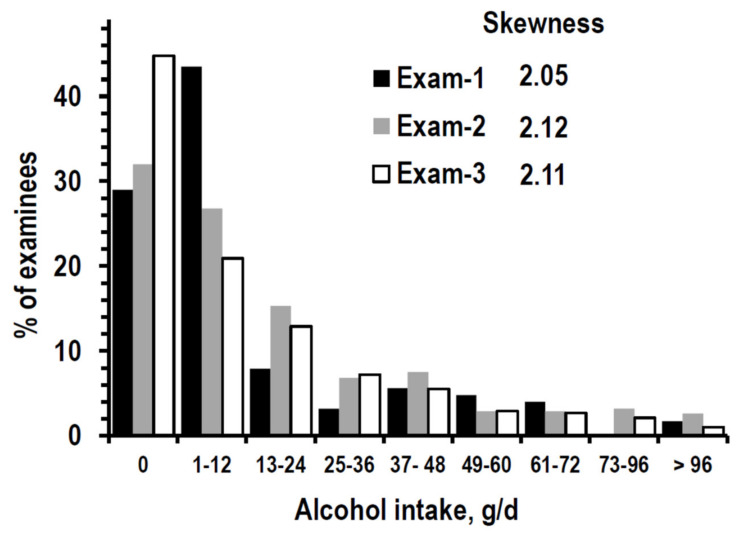
Skewness and frequency distribution of alcohol intake (g/d = g/day) at Exam-1 (black bars), Exam-2 (grey bars), and Exam-3 (white bars) in the 2069 examinees with complete data at all exams.

**Figure 3 nutrients-14-01297-f003:**
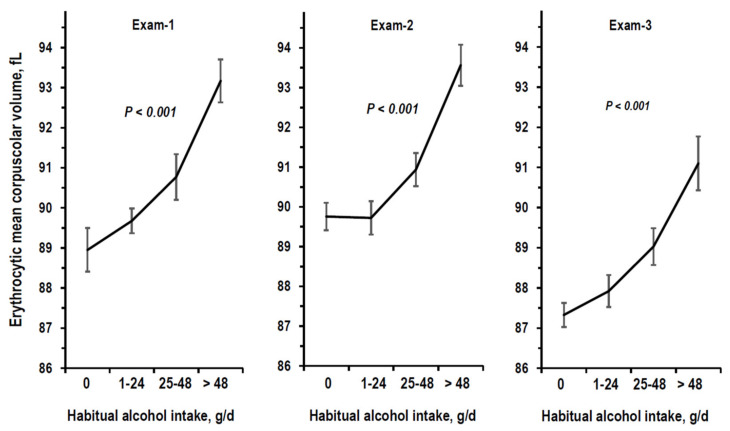
Mean and 95%CI of erythrocytic mean corpuscular volume by stratum of alcohol intake (g/d = g/day) at Exam-1, Exam-2, and Exam-3. Number of examinees per stratum is reported in Table 1. *p*-values are from non-adjusted ANOVA.

**Figure 4 nutrients-14-01297-f004:**
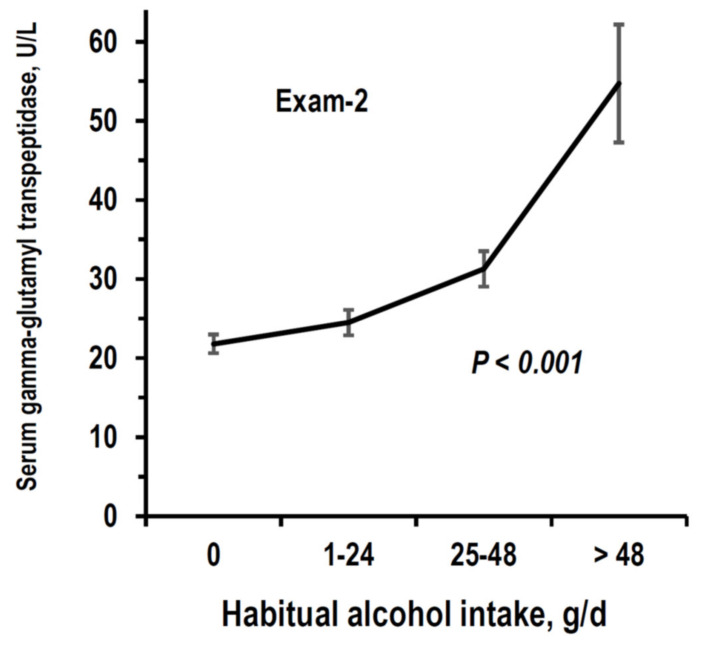
Mean and 95%CI of serum gamma-glutamyl transpeptidase by stratum of alcohol intake (g/d = g/day) at Exam-2. Number of examinees per stratum is reported in Table 1. *p*-value is from non-adjusted ANOVA.

**Figure 5 nutrients-14-01297-f005:**
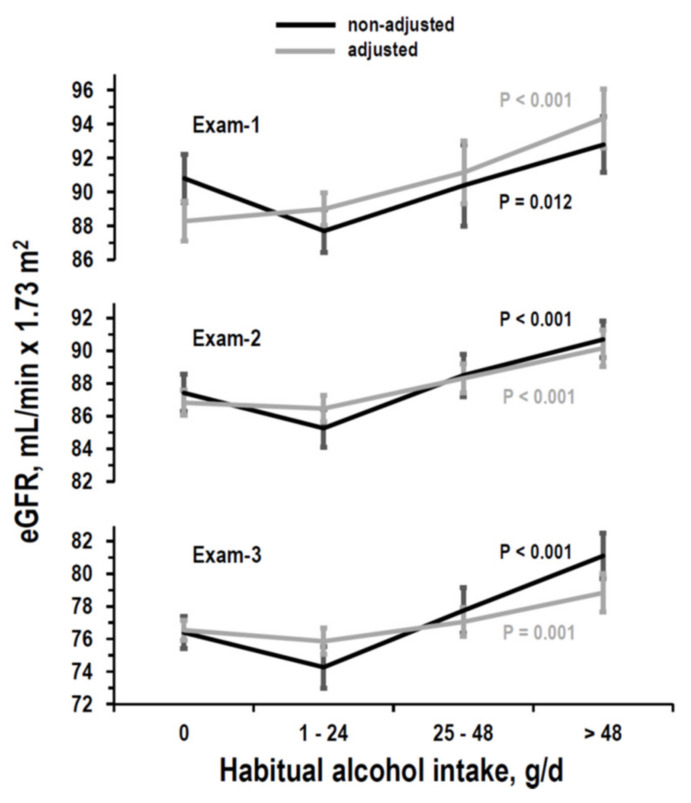
Cross-sectional analyses: mean and 95%CI of eGFR by stratum of alcohol intake (g/d = g/day) at Exam-1, Exam-2, and Exam-3 in non-adjusted ANOVA (black lines) and ANOVA adjusted for covariates (grey lines). Number of examinees per stratum is in Table 1. *p*-values are from ANOVA. Covariates in adjusted ANOVA for Exam-1 data: gender and data at Exam-1 for age, education, log-transformed urinary sodium/creatinine ratio, log-transformed urinary potassium/creatinine ratio, urinary creatinine, body mass index, systolic pressure, diastolic pressure, antihypertensive drug treatment, serum total cholesterol, smoking, and diabetes. Covariates in adjusted ANOVA for Exam-2 data: gender and data at Exam-2 for age, education, log-transformed urinary sodium/creatinine ratio, log-transformed urinary potassium/creatinine ratio, log-transformed urinary urea nitrogen/creatinine ratio, urinary creatinine, body mass index, systolic pressure, diastolic pressure, antihypertensive drug treatment, serum total cholesterol, smoking, and diabetes. Covariates in adjusted ANOVA for Exam-3 data: gender and data at Exam-3 for age, education, urinary creatinine, body mass index, systolic pressure, diastolic pressure, antihypertensive drug treatment, serum total cholesterol, smoking, and diabetes.

**Figure 6 nutrients-14-01297-f006:**
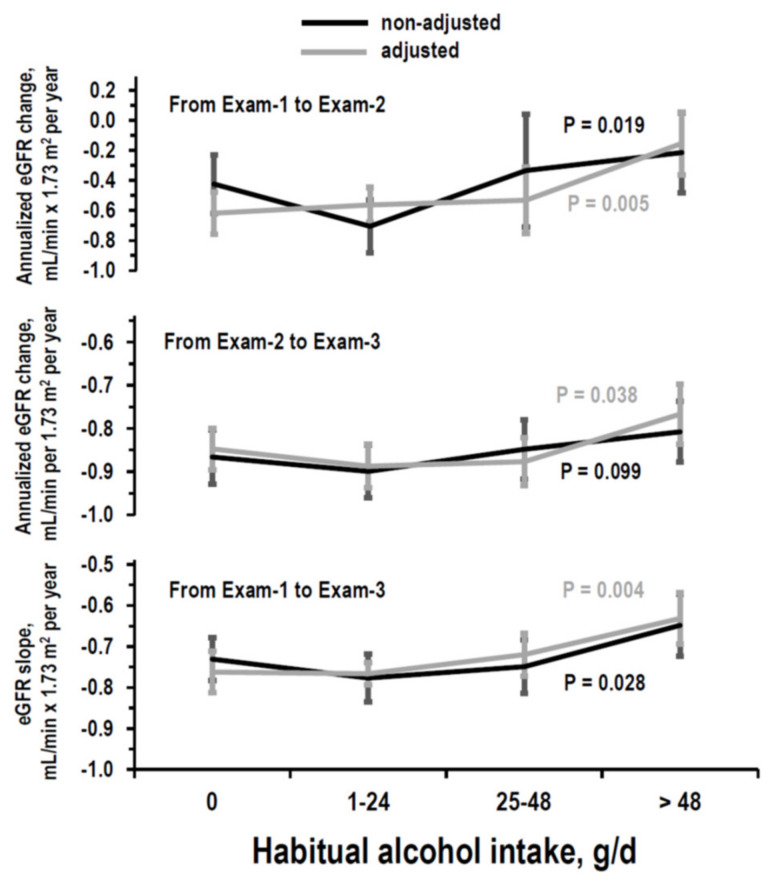
Longitudinal analyses: mean and 95%CI of annualized eGFR change from Exam-1 to Exam-2 by stratum of alcohol intake (g/d = g/day) at Exam-1, of annualized eGFR change from Exam-2 to Exam-3 by stratum of alcohol intake at Exam-2, and of eGFR slope over time from Exam-1 to Exam-3 by stratum of mean alcohol intake during follow-up in non-adjusted ANOVA (black lines) and in ANOVA adjusted for covariates (grey lines). *p*-values are from ANOVA. Number of examinees per alcohol stratum is in Table 1 for eGFR change from Exam-1 to Exam2 and for eGFR change from Exam-2 to Exam3. Number of examinees per alcohol stratum for eGFR slope is as follows: 0 g/day = 331, 1–24 g/day = 1146, 25–48 g/day = 331, and > 48 g/day = 261. Covariates in adjusted ANOVA on eGFR change from Exam-1 to Exam-2: gender and data at Exam-1 for age, eGFR, education, log-transformed urinary sodium/creatinine ratio, log-transformed urinary potassium/creatinine ratio, urinary creatinine, body mass index, systolic pressure, diastolic pressure, antihypertensive drug treatment, serum total cholesterol, smoking, and diabetes. Covariates in adjusted ANOVA on eGFR change from Exam-2 to Exam-3: gender and data at Exam-2 for age, eGFR, education, log-transformed urinary sodium/creatinine ratio, log-transformed urinary potassium/creatinine ratio, log-transformed urinary urea nitrogen/creatinine ratio, urinary creatinine, body mass index, systolic pressure, diastolic pressure, antihypertensive drug treatment, serum total cholesterol, smoking, and diabetes. Covariates in adjusted ANOVA on eGFR slope over time from Exam-1 to Exam-3: gender, data at Exam-1 for age, eGFR, antihypertensive drug treatment, smoking, and diabetes, and means of data available from Exam-1 to Exam-3 for education, log-transformed urinary sodium/creatinine ratio (not measured at Exam-3), log-transformed urinary potassium/creatinine ratio (not measured at Exam-3), log-transformed urinary urea nitrogen/creatinine ratio (measured at Exam-2 only), urinary creatinine, body mass index, systolic pressure, diastolic pressure, and serum total cholesterol.

**Table 1 nutrients-14-01297-t001:** Descriptive statistics in examinees with complete data at Exam-1, Exam-2, and Exam-3: prevalence for categorical variable, mean ± SD for non-skewed variables, and median (IQR) for skewed variables.

		Exam-1	Exam-2	Exam-3
Date of exam, year		1983–1985	1989–1992	2001–2007
Number of examinees		2069	2069	2069
% men		42.8%	42.8%	42.8%
Age, years		43 ± 13	49 ± 13	62 ± 12
Alcohol intake	g/day	12 (0/25)	12 (0/36)	12 (0/24)
	% reporting 0 g/day	29.0% (601)	32.0% (663)	44.8% (927)
	% reporting 1–24 g/day	43.5% (899)	27.3% (565)	22.8% (472)
	% reporting 25–48 g/day	11.1% (230)	21.0% (454)	19.4% (401)
	% reporting > 48 g/day	16.4% (339)	18.7% (387)	13.0% (269)
eGFR, mL/min × 1.73 m^2^		91 ± 18	88 ± 14	77 ± 14
**Covariates**				
Urinary sodium/creatinine, mmol/g		107 (74/151)	103 (68/145)	not assessed
Urinary potassium/creatinine, mmol/g		29 (22/38)	24 (19/33)	not assessed
Urinary urea nitrogen/creatinine, g/g		not assessed	9.9 (7.6/12.1)	not assessed
Education, year		7.6 ± 4.3	7.9 ± 4.4	not assessed
Urinary creatinine, g/24-h		1.26 ± 0.31	1.25 ± 0.31	1.18 ± 0.33
Body mass index, kg/m^2^		26.1 ± 4.2	26.8 ± 4.2	27.2 ± 4.3
Systolic pressure, mm Hg		127 ± 18	125 ± 18	134 ± 19
Diastolic pressure, mm Hg		76 ± 11	75 ± 10	77 ± 10
On antihypertensive drug, % (*n*)		8.8% (183)	14.8% (306)	43.7% (905)
Serum total cholesterol, mg/dL		206 ± 44	219 ± 40	216 ± 38
Smoking, % (*n*)		15.6% (322)	31.7% (656)	22.2% (459)
Diabetes, % (*n*)		1.1% (22)	3.2% (67)	8.2% (170)

**Table 2 nutrients-14-01297-t002:** Descriptive statistics in 2069 examinees with complete data at all exams for follow-up duration and eGFR data (mean ± SD).

Number of Examinees		2069
Men, *n* (%)		42.8% (886)
Follow-up duration, years	from Exam-1 to Exam-2	5.94 ± 0.97
	from Exam-2 to Exam-3	13.34 ± 2.08
	from Exam-1 to Exam-3	19.3 ± 2.05
Annualized eGFR change *	from Exam-1 to Exam-2	−0.51 ± 0.84
	from Exam-2 to Exam-3	−0.85 ± 0.76
eGFR slope *	from Exam-1 to Exam-3	−0.74 ± 0.71

* units = mL/min × 1.7 m^2^ per year.

**Table 3 nutrients-14-01297-t003:** Cross-sectional analysis: multi-variable linear regression models for data of Exam-1, Exam-2, and Exam-3 with eGFR regressed over stratum of alcohol intake.

		Dependent Variable		
		Exam-1 eGFR	Exam-2 eGFR	Exam-3 eGFR
		Model 1	Model 2	Model 3
**Habitual alcohol intake, g/day**	0 (non-drinker)	0 (reference)	0 (reference)	0 (reference)
	1–24	B = 0.707 (*−1.53/2.94*) *p* = 0.535	B = −0.360 (*−1.45/0.73*) *p* = 0.515	B = −0.686 (*−1.68/0.31*) *p* = 0.177
	25–48	B = 2.869 (*1.36/4.38*) *p* < 0.001	B = 1.491 (*0.30/2.68*) *p* = 0.014	B = 0.505 (*−0.61/1.62*) *p* = 0.376
	>48	B = 6.056 (*3.87/8.24*) *p* < 0.001	B = 3.336 (*1.89/4.78*) *p* < 0.001	B = 2.284 (*0.88/3.69*) *p* = 0.001

Regression coefficient (B), 95% confidence interval (*italic*), and *p*-value. Covariates included in Model 1: gender and data at Exam-1 for age, education, log-transformed urinary sodium/creatinine ratio, log-transformed urinary potassium/creatinine ratio, urinary creatinine, body mass index, systolic pressure, diastolic pressure, antihypertensive drug treatment, serum total cholesterol, smoking, and diabetes. Covariates included in Model 2: gender and data at Exam-2 for age, education, log-transformed urinary sodium/creatinine ratio, log-transformed urinary potassium/creatinine ratio, log-transformed urinary urea nitrogen/creatinine ratio, urinary creatinine, body mass index, systolic pressure, diastolic pressure, antihypertensive drug treatment, serum total cholesterol, smoking, and diabetes. Covariates included in Model 3: gender and data at Exam-3 for age, education, urinary creatinine, body mass index, systolic pressure, diastolic pressure, antihypertensive drug treatment, serum total cholesterol, smoking, and diabetes. Number of examinees per stratum of alcohol intake is the same as shown in Table 1.

**Table 4 nutrients-14-01297-t004:** Longitudinal analyses: multi-variable linear regression models with annualized eGFR change and eGFR slope regressed over stratum of alcohol intake and covariates.

		Dependent Variable		
		Annualized eGFR Change from Exam-1 to Exam-2	Annualized eGFR Change from Exam-2 to Exam-3	eGFR Slope from Exam-1 to Exam-3
		Model 1	Model 2	Model 3
**Habitual alcohol intake, g/day**	0 (non-drinker)	0 (reference)	0 (reference)	0 (reference)
	1–24	B = 0.055 (*−0.13/0.24*) *p* = 0.549	B = −0.040 (*−0.11/0.03*) *p* = 0.249	B = −0.006 (*−0.06/0.05*) *p* = 0.841
	25–48	B = 0.086 (*−0.18/0.35*) *p* = 0.525	B = −0.029 (*−0.10/0.05*) *p* = 0.437	B = 0.037 (*−0.04/0.11*) *p* = 0.336
	>48	B = 0.464 (*0.20/0.73*) *p* = 0.001	B = 0.158 (*0.07/0.25*) *p* = 0.016	B = 0.136 (*0.05/0.22*) *p* = 0.002

Regression coefficient (B), 95% confidence interval (*italic*), and *p*-value. Covariates included in Model 1 = gender and data at Exam-1 for age, eGFR, education, log-transformed urinary sodium/creatinine ratio, log-transformed urinary potassium/creatinine ratio, urinary creatinine, body mass index, systolic pressure, diastolic pressure, antihypertensive drug treatment, serum total cholesterol, smoking, and diabetes. Covariates included in Model 2 = gender and data at Exam-2 for age, eGFR, education, log-transformed urinary sodium/creatinine ratio, log-transformed urinary potassium/creatinine ratio, log-transformed urinary urea nitrogen/creatinine ratio, urinary creatinine, body mass index, systolic pressure, diastolic pressure, antihypertensive drug treatment, serum total cholesterol, smoking, and diabetes. Covariates included in Model 3 = gender, age, eGFR, antihypertensive drug treatment, smoking, and diabetes at Exam-1 and mean data available from Exam-1 to Exam-3 for education, log-transformed urinary sodium/creatinine ratio, log-transformed urinary potassium/creatinine ratio, log-transformed urinary urea nitrogen/creatinine ratio, urinary creatinine, body mass index, systolic pressure, diastolic pressure, serum total cholesterol. Number of examinees per stratum of alcohol intake for Model 1 and Model 2 is in Table 1. Number of examinees per stratum of alcohol intake for model 3 is as follows: 0 g/day = 331, 1–24 g/day = 1146, 25–48 g/day = 331, and >48 g/day = 261.

**Table 5 nutrients-14-01297-t005:** Analysis of mortality: multi-variable Cox regression models with mortality rate during follow-up, regressed over stratum of alcohol intake and covariates at Exam-1, in all examinees with complete data and subgroups with decreased eGFR.

		All Exam-1 Participants	Exam-1 Participants with eGFR = 89–60 mL/min × 1.73 m^2^	Exam-1 Participants with eGFR < 60 mL/min × 1.73 m^2^
**Number of examinees**		4524	2201	392
**eGFR range, mL/min × 1.73 m^2^**		195–21	89–60	59–21
**Number of deaths**		992	609	214
**Patients years product**		91832	43241	6208
**Habitual alcohol intake, g/day**	0 (non-drinker)	1 (reference)	1 (reference)	1 (reference)
	1–24	HR = 0.767 (*0.64/0.91*) *p* = 0.003	HR = 0.820 (*0.65/1.03*) *p* = 0.090	HR = 0.687 (*0.49/0.97*) *p* = 0.033
	25–48	HR = 1.100 (*0.88/1.37*) *p* = 0.394	HR = 1.24 (*0.95/1.63*) 0.112	HR = 0.802 (*0.44/1.46*) *p* = 0.0.941
	>48	HR = 0.916 (*0.74/1.14*) *p* = 0.431	HR = 0.889 (*0.67/1.17*) *p* = 0.405	HR = 0.973 (*0.48/1.99*) *p* = 0.941

Hazard ratio (HR), 95% confidence interval (*italic*), and *p*-value. Covariates included in all models: gender and data at Exam-1 for age, eGFR, education, log-transformed urinary sodium/creatinine ratio, log-transformed urinary potassium/creatinine ratio, urinary creatinine, body mass index, systolic pressure, diastolic pressure, antihypertensive drug treatment, serum total cholesterol, smoking, and diabetes.

## Data Availability

Data are stored in the Gubbio study repository. For information, write to mlaurenzi@comcast.net.

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
