# Peer review of "Relation of Alcohol Intake to Kidney Function and Mortality Observational, Population-Based, Cohort Study"

_nutrients, 2022, doi:10.3390/nu14061297_

Round 1

Reviewer 1 Report

In this Italian cross-sectional and follow-up studies, authors showed that alcohol intake showed positive relation to eGFR at all exams, i.e. Exam-1(baseline), Exam-2(6-year) and Exam-3(20-year), cross-sectionally; and related longitudinally to less negative eGFR change from Exam-1 to Exam-2 and from Exam-2 to Exam-3. Some of the results are interesting, however, there are several concerns to be addressed.

1. The risk of death and related disease development is usually higher among abstainers. In this study, abstainers are considered to be in the 0-gram group; is this correct? In this case, the risk of the drinking group may be lower because of the increased risk of the 0-gram group. It may be necessary to conduct an analysis that excludes those who newly moved to the 0-gram group during the follow-up period, for example.

2. Blood pressure usually increases with increased alcohol consumption. Hypertension is a major cause of impaired renal function. What was the association between drinking strata and blood pressure in this study? Are the results of the present study equally acceptable when divided by, for example, median of systolic blood pressure at baseline survey? And how was information on antihypertensive medication during the follow-up period handled?

3. In Supplementary Table S4, author only showed median of alcohol intake of examinees participating in all exams, in examinees dead during follow-up, and in examinees not dead and lost to follow-up. However, important information is proportions of 4 strata of drinking categories, which may make a bias in their findings. Show that information and discuss.

Reviewer 2 Report

The article “Relation of Alcohol Intake to Kidney Function and Mortality Observational, Population-based, Cohort Study” by M. Cirillo and colleagues

The article is well-presented and detailed. In my opinion, no major issues are present and, despite the obvious limits of the study because of the many variables involved (as also stated by the authors).

However, a couple of minor issues (as follows) would improve the quality of the study.

Inclusion of a scheme depicting the overall project, e.g. with stages and exams, would also help the reader to better and more easily understand the study.

More technical details must be provided in section 2 (Materials and methods), e.g. number of participants in total and in each group, age, sex, weight, etc (some of these data are already present in Table 1 and should be removed from there). Inclusion of these data in a table would help.
